# Single-Strand Annealing in Cancer

**DOI:** 10.3390/ijms22042167

**Published:** 2021-02-22

**Authors:** Janusz Blasiak

**Affiliations:** Department of Molecular Genetics, Faculty of Biology and Environmental Protection, University of Lodz, 90-236 Lodz, Poland; janusz.blasiak@biol.uni.lodz.pl; Tel.: +48-426354334

**Keywords:** single-strand annealing, DNA double-strand break repair, SSA, homologous recombination, cancer, RAD52, synthetic lethality, BRCAness, therapeutic genome editing, CRISPR/Cas9

## Abstract

DNA double-strand breaks (DSBs) are among the most serious forms of DNA damage. In humans, DSBs are repaired mainly by non-homologous end joining (NHEJ) and homologous recombination repair (HRR). Single-strand annealing (SSA), another DSB repair system, uses homologous repeats flanking a DSB to join DNA ends and is error-prone, as it removes DNA fragments between repeats along with one repeat. Many DNA deletions observed in cancer cells display homology at breakpoint junctions, suggesting the involvement of SSA. When multiple DSBs occur in different chromosomes, SSA may result in chromosomal translocations, essential in the pathogenesis of many cancers. Inhibition of RAD52 (RAD52 Homolog, DNA Repair Protein), the master regulator of SSA, results in decreased proliferation of BRCA1/2 (BRCA1/2 DNA Repair Associated)-deficient cells, occurring in many hereditary breast and ovarian cancer cases. Therefore, RAD52 may be targeted in synthetic lethality in cancer. SSA may modulate the response to platinum-based anticancer drugs and radiation. SSA may increase the efficacy of the CRISPR (Clustered Regularly Interspaced Short Palindromic Repeats)/Cas9 (CRISPR associated 9) genome editing and reduce its off-target effect. Several basic problems associated with SSA, including its evolutionary role, interplay with HRR and NHEJ and should be addressed to better understand its role in cancer pathogenesis and therapy.

## 1. Introduction

DNA double-strand breaks (DSBs) are, along with interstrand-crosslinks (ICLs), the most serious kind of DNA damage, and may result in chromosome breaks leading to chromosomal translocation, production of fusion genes, often displaying oncogenic properties to fuel cancer transformation [1]. DSBs can be induced by variety of environmental factors, including ionizing radiation, radiomimetic chemicals, and anticancer drugs (reviewed in [2]). DSBs may also result from two independent DNA single-strand breaks (SSBs) located on two DNA strands in a close vicinity–the exact distance between two SSBs to produce a DSB depends on the DNA base composition and its architecture within the nucleus [3]. The possibility of the conversion of two SSBs into one DSB significantly extends the range of factors that may induce DSBs. However, DSBs may also result from disturbed DNA replication, transcription, and recombination, and can be an intermediate in ICL repair or V(D)J (variable(diversity)joining) recombination in immunoglobins and T-cell receptor maturation [4]. Programmed DSBs are induced by the Spo11 (SPO11 Initiator of Meiotic Double Stranded Breaks) endonuclease in meiosis [5].

As DSBs are a direct threat to the cell, the cell has evolved pathways to deal with such a danger. DNA damage response (DDR), a complex multi-protein, evolutionary conserved cellular reaction to DNA damage is best known in the context of DSBs challenge (reviewed in [6]). It is initiated by the activation of the ATM (ATM Serine/Threonine Kinase), ATR (ATR Serine/Threonine Kinase), and DNA-PK_CS_ (Protein Kinase, DNA-Activated, Catalytic Subunit) to phosphorylate multiple substrates [7]. The execution stage of DSB-induced DDR includes several effects, comprising chromatin remodeling, cell cycle modulation, gene expression, and DSB repair (DSBR). In general, two mechanisms of DSBR can be considered: those based on homology (homology-directed repair, HDR) and those in principle independent of homology, but which occasionally make use of it. Humans and other higher organisms repair DSBs primarily by non-homologous end joining (NHEJ) and homologous recombination repair (HRR). Another DSBR pathway is single-strand annealing (SSA), but it seems to be of a minor significance compared with NHEJ and HRR (reviewed in [8]). Homologous recombination repair may proceed according to the classical model of homologous recombination (HR) based on DSB repair and double Holliday junction resolution or take the form of SDSA (synthesis-dependent strand annealing) or BIR (break-induced replication). In turn, at least two variants of NHEJ can be considered: C-NHEJ (canonical NHEJ) and B-NHEJ (backup NHEJ), including MMEJ (microhomology-mediated end joining) (Figure 1).

HRR in this review is understood as HR resulting in double Holliday junction resolution.

Since genomic instability is typical for most, if not all, cancers, DDR is essential in the inhibition of cancer transformation in all its stages [9]. The roles of HRR and NHEJ in cancer have been addressed in many studies, but SSA is not often considered in cancer-related research (reviewed in [10,11]). In this review, a short overview of the SSA mechanism and its role in DDR, DSBR, and genome maintenance is presented. Also, updated information on the involvement of SSA in cancer transformation and therapy is presented. 

## 2. Single Strand Annealing–A Distinct Pathway in DNA Double-Strand Break Repair

Despite some mechanistic similarities, SSA should not be considered an HRR or NHEJ variant, as its functioning is underlined by different proteins and biochemical reactions [12]. Al-Minawi et al., showed that SSA was not influenced by inhibition of essential proteins of DDR: ATM, CHK1 (Checkpoint Kinase 1), CDK2 (Cyclin Dependent Kinase 2), or DNA-PK (DNA-dependent Protein Kinase) [13]. 

As presented in Figure 1, end resection is a crucial mechanism in the choice between C-NHEJ and HDR. However, HDR includes several pathways; thus, the question is whether end resection may be involved in discrimination between HDR and B-NHEJ. Initial, relatively short resection is a signal for RAD51 (RAD51 Double Strand Break Repair Protein) to form a nucleofilament with ssDNA overhang, resulting from the resection of the complimentary strand. HRR, SDSA, and BIR are initiated by strand invasion, which is led by the RAD51 nucleofilament. Neither SSA nor MMEJ require strand invasion. However, strand invasion may be effective only when a donor DNA is juxtaposed, but the presence of donor DNA is not enough when there is no functional RAD51, as in the case of BRCA mutants (BRCAness) [8]. The strand invasion, led by RAD51, is facilitated by the two distinct sites in the RAD51 filament: primary, accommodating ssDNA, and secondary, which can accommodate double dsDNA in a weak, transient fashion, independent of DNA sequence. In this way, RAD51 may check long DNA stretches for homology. When RAD51 finds it, primary and secondary sites exchange ssDNA, forming new duplexes and promoting a reciprocal invasion of the template on the damaged DNA. In general, ablation of RAD51 stimulates SSA, but does not activate C-NHEJ. (Figure 1). It is possible that if the repeats nearby to DSB are short (4-5 bp), MMEJ is preferred, and when they are longer, SSA may be activated. Moreover, the distance between DSB and nearby repeats may be important. Furthermore, it was shown that repressing SSA and MMEJ by RAD51 resulted from its non-catalytic binding to ssDNA and was independent of its invasion/strand exchange catalytic activity [1].

In general, SSA is a DSBR pathway occurring when at least two direct DNA repeats occur on the two ends of a DSB (Figure 2). Therefore, in contrast to HRR, SSA does not need a donor sequence. However, a subset of such repeats can be a substrate for B-NHEJ/MMEJ. The regulation of the choice between DSBR pathways is not fully known. In the context of SSA choice it seems important to consider the resection step, as it also occurs in HRR and B-NHEJ. So et al., and others suggested that the choice may follow two steps: discrimination between end resection and C-NHEJ, and competition among HRR, B-NHEJ, and SSA [14,15,16]. 

Although not all details of DSBR choice are fully known, the resection step is its major decisive process. At the initial stage, DNA ends are resected by the interaction between the structure-specific nucleases MRE11 (MRE11 Homolog, Double Strand Break Repair Nuclease) and CtIP (RB Binding Protein 8, Endonuclease) [17,18]. MRE11 is a component of the MRN (MRE11, RAD50 (RAD50 Double Strand Break Repair Protein) and NBS1 (Nibrin)) complex, which binds DNA through the association of MRE11 and NBS1 with the A and B Walker motifs in RAD50 and the interaction with the extended coiled-coil tail of RAD50 [19]. These interactions allow the MRN complex to form a bridge between DNA ends. CtIP and ATM dimers are then recruited to the DSB site by specific domains in NBS1. CtIP functions together with MRN, but its nuclease activity is likely required to process complex DNA ends to enable the action of end-resecting exonucleases that need 3-OH and 5′-P ends [20]. As MRE11 has 3′-5′ exonucleolytic activity, opposite to 5′-3′ counterpart required for 3′-overhangs formation in end resection, it firstly induces a nick in DNA using its endonucleolytic activity and then degrades DNA towards DSB, producing 3′ ssDNA tails [21]. Such resected DNA ends ranging about 100 nt may be a substrate for MMEJ, a variant of B-NHEJ [22]. Longer single-strand stretches (extensive resection) are generated by the combined action of helicases and endonucleases, including DNA2 (DNA Replication Helicase/Nuclease 2), BLM (BLM RecQ like helicase), WRN (WRN RecQ Like Helicase), CtIP, and EXO1 (Exonuclease 1) [23,24].The cell cycle phase is a major factor to choose between HRR and SSA and the process of end resection is supported by cyclin-dependent kinases (CDKs) [25]. CDKs may phosphorylate CtIP facilitating its interaction with BRCA1 (BRCA1 DNA Repair Associated) in S and G2 phases [26]. However, later it was shown that CtIP might act independently of BRCA1 in the resecting reaction [27]. Recent data suggest that BRCA1 accelerates CtIP-mediated end resection and plays an important role in the choice of DSBR [22,28]. CtIP is prevented from binding DNA ends by TP53BP1 (Tumor Protein P53 Binding Protein 1), which directs DSBR machinery to C-NHEJ [29]. ATM, an essential regulator of DDR and HDR, is also speculated to be involved in end resection and DSBR choice [30]. 

Resected long 3′-ss ends are targeted by RPA (Replication Protein A) and RAD51 [31]. RPA first binds ssDNA and is then displaced through the action of mediators, including BRCA2 and PALB2 (Partner and Localizer of BRCA2) to allow RAD51 filament formation. Therefore, in the absence of functional RAD51, as is the case in BRCA mutants, RPA binding to 3-ss ends prevents annealing between short homologies, avoids MMEJ and supports SSA, which is independent of RAD51 as it requires neither donor sequence nor strand invasion [32]. End resection allows SSA to recognize complementary sequences on both DNA ends to proceed with their annealing, which is mediated by RAD52 (RAD52 Double Strand Break Repair Protein). Non-homologous 3′-ss tails, protruding from dsDNA structure, are targeted by ERCC1 (ERCC Excision Repair 1, Endonuclease Non-Catalytic Subunit), which associates with XPF (ERCC4, ERCC Excision Repair 4, Endonuclease Catalytic Subunit) to form a complex, supported by RAD52, to cleave 3′-ss DNA tails [33]. Any gaps are then filled by DNA polymerases and DNA ligase completes the process of SSA. However, not all polymerases/ligases that may be important for SSA have been identified and characterized [34]. Recently, it was shown that RAD52 activity is enhanced by its direct, high affinity and 1:1 stoichiometry binding to the highly conserved, small, and very acidic protein called DSS1 (SEM1, 26S Proteasome Complex Subunit) [35]. Such interaction changed the conformation of RAD52 and modulated its binding to DNA and was important for SSA as it increased the efficacy of the SSA reaction four-fold. Further research showed that that effect was mainly attributed to the initial rate of RAD52 ssDNA annealing. The DSS1-RAD52 complex was also shown to be important for BIR. 

Therefore, SSA causes deletion of one repeat sequence and the intervening sequences between repeats, resulting in deletion and copy number alterations. The end resection process and lack of an intact template resulting in a deletion cause SSA of a potentially mutagenic DSBR. Therefore, an immediate question arises: what is responsible for the choice of this system of DNA repair over largely non-mutagenic HRR [34]? Or, conversely, is there any cellular evolutionary mechanism suppressing this system from being too active?

Scully et al., considered a state when two DNA ends of a tandem duplication were repaired by SSA so the duplicated segments overlapped each other, changing the duplication into an original single copy region [8]. In those circumstances, SSA would accomplish a conservation at a stalled replication fork by preventing the formation of tandem repeats by abnormally restarted replication fork. Therefore, error-prone SSA may hypothetically turn into error-free repair at a stalled replication fork.

Another important question is why SSA is needed at all, if it is mutagenic and the cell has a potentially error-free DSBR pathway, HRR? This question is similar to the question about NHEJ and its answer is also like the answer in the NHEJ case—the cellular context may decide on the choice of repair pathway. HRR is a multi-protein and a complex DNA repair system requiring considerable time that not always can be given, sometimes a quick action is needed to support cellular functions. Cellular context is strictly associated with the cell cycle—HRR requires elongated DNA fragments and preferentially uses sister chromatids to carry out the repair. End resection does not require prior synthesis of sister chromatids [17]. Therefore, when DSB occurs in the S phase in the absence of sister chromatids, it may be repaired by SSA or MMEJ due to lack of a preferred template for HRR [36]. Yet another cellular context influencing the choice between SSA and HRR is the state of chromatin, which determines DNA damage accessibility. More closed chromatin would impede the building of a multi-protein complex required for HRR at the damage site, while SSA proteins may be compatible with such configuration. In general, the chromatin state is determined by the cell cycle phase, but it may also change due to external stimulus, which may be the case of the action of DNA-damaging factors, as they affect not only DNA but proteins as well. 

Both SSA and MMEJ anneal homologous sequences of DNA to bridge its two broken ends. Moreover, most DDR factors that promote MMEJ also promote SSA [37]. The main differences between these DSBRs lie in (1) the length of 3′-ss DNA tails: short in MMEJ versus extensive in SSA, (2) the length of annealing intermediate: very short in MMEJ versus extensive stretch of interrupted homology in SSA and (3) mediators of synapsis: PARP (Poly-ADP-ribose Polymerase) and Polθ in MMEJ versus RAD52 in SSA [34]. However, (2) might not clearly determine the choice between SSA and MMEJ when a homology of intermediate length occurs. Is there a sharp boundary separating homology lengths specific for these systems?

There are also similarities between SSA and HRR. In fact, in some cases it is practically impossible to distinguish between the outcomes of these two DSBRs. One such case is HRR, with the use of two tandem repeats and resolving with crossing over [38]. The main distinction between these two DSBRs is the use of RAD51 by HRR. SSA does not require RAD51, and, moreover, disruption of RAD51 was reported to increase SSA [34].

In *Saccharomyces cerevisiae*, branched intermediates in SSA are stabilized by mismatch repair (MMR) protein Msh2 (DNA Mismatch Repair Protein MSH2) and Msh3 [39]. Other MMR proteins, including Msh6, may be important to suppress SSA for non-identical (homeologous) repeats in a process called heteroduplex rejection [40,41]. A suppressed recombination between homologous sequences was shown in studies with mouse and human cells with mutations in the *MSH2/MSH3/MSH6* genes [42,43,44]. Later, it was shown that the products of the mammalian *MLH1*, *PMS2* (PMS1 Homolog 2, Mismatch Repair System Component), and *MLH3* genes suppressed SSA of heterologous repeats in mouse cells [45]. However, Howard et al., showed that MSH6 promoted SSA in human bone osteosarcoma epithelial U2OS cell line [37].

## 3. SSA in Genomic Instability and DNA Repair Defects in Cancer

Genomic instability, underlined by defects in DNA repair, is a hallmark of most, if not all, cancers and promotes the evolution and heterogeneity of cancer cells [46]. Therefore, defects in the DNA repair pathway may be essential for cancer initiation and progression. Moreover, as SSA is per se mutagenic, even its normal functioning to repair DSBs may result in deletions in genes important for cancer transformation. SSA-specific deletions were observed between homologous segments of repeats in germ-line mutations of several tumor suppressor genes [16].

DNA DSBs are usually pictured as two single-strand breaks (SSBs) located opposite each other on two complimentary DNA strands. However, DSBs can occur in many other forms to be a substrate for DSBR. Structurally, DSBs can be categorized into two classes—one-ended and two-ended (Figure 3A). However, independently of class, DSBs rarely have canonical 3′-OH or 5′-P termini—instead, they have complex chemical groups or adopt a hairpin structure [47]. Therefore, they must be proceeded before end resection or direct ligation. In canonical SSA in normal conditions, DNA repeats are in the simple form, i.e., they have the same sequence when being read in the same direction on either side of DSB. However, several reports, mostly in yeast, suggest that inverted DNA repeats can also be a substrate for SSA [48,49,50,51] (Figure 3B). This fact has potential significance for cancer pathogenesis as DNA palindromes are reported to colocalize in cancer cells with chromosomal regions that are predisposed to gene amplification, an important mechanism in etiology of many cancers [52].

The general conclusion from research on the role of inverted repeats in SSA is that they promote unusual DNA structures, including inverted chromosomal dimers that can be in a dicentric form as well as fold-backs. Ramakrishan et al., showed that inverted repeats could form two types of unusual DNA structures that are determined by the length between inverted repeats [50]. When the repeats were separated by a long (about 1 kb) spacer, they were a substrate for intermolecular SSA resulting in the formation of inverted dimers. However, when the distance between repeats was 12 bp, they were a substrate for intra-molecular SSA, producing fold-back structures. Both these DNA forms channeled DSBR into BIR, a genome destabilizing DSBR pathway. Therefore, inverted repeats, when subjected by SSA, may lead to genome instability, essential for cancer induction and progression. However, most studies on SSA and inverted repeats were performed in yeast and their validation in mammalian cells is needed. 

It was reported that precancerous and cancerous-associated chronic expression of p21 (*CDKN1A*, Cyclin Dependent Kinase Inhibitor 1A) in p53-deficient cells induced a deleterious form of replication stress in a senescence-like phase with error-prone DNA repair [53]. Further, it was shown that in such an environment a chronic p21 expression independent of p53 resulted in increased DSBs, resulting from their repair by RAD52-dependent BIR and SSA [54]. Moreover, these studies showed that RAD52 was activated transcriptionally in a E2F1 (E2F Transcription Factor 1)-dependent fashion. In summary, RAD52 and SSA may belong to the main sources of genomic instability, vital for cancer induction and development, so they can be considered in anticancer strategies. 

Sirtuin 1 (SIRT1) was shown to mediate HRR with the involvement of WRN, linked to SSA [55]. A database of 596 DNA repair genes was constructed and analyzed in the cases of cervical cancer induced by infection of HPV16 (human papillomavirus) and HPV18 strains [56]. It was noted that HPV18+ patients who had poor prognosis after postoperative radiation therapy showed more effective DNA repair processes that were underlined by several genes and DNA repair pathways, including *SIRT1* and SSA.

Suppression of RAD51 in Barret’s adenocarcinoma cells caused activation of SSA, increasing genomic instability in these cells [57].

In summary, SSA could contribute to cancer-related genomic instability even in its canonical form. It seems especially important when DSBR is switched from an error-free pathway to SSA, as such a switch usually occurs in non-physiological conditions favoring non-canonical, error-prone SSA, further increasing genomic instability.

DNA repair defects underlined by BRCA1/2 deficiency and chromosomal translocations will be considered in the subsequent sections.

## 4. SSA in Chromosomal Rearrangements in Cancer

Chromosomal translocations occur when DNA ends of DSBs on heterologous chromosomes are joined. In general, effective action of SSA in a proper cellular context contributes to genomic stability, but its faulty activity may lead to chromosomal rearrangements that increase genomic instability [10]. It is especially important in the human genome, consisting of individual chromosomes with a substantial proportion of repeated sequences. Of note, many mutagenic deletions linked with cancer and human genetic diseases show homology at breakpoint junctions, suggesting the involvement of SSA in such effects. When DSBs occurring in two heterologous chromosomes are flanked by the same repeats and these chromosomes are in proximity to each other, SSA may erroneously anneal single strands of these chromosomes, which may result in a chromosome translocation (Figure 4). However, annealing of single strands from different chromosomes does not necessarily imply chromosome fusion and this effect requires further studies.

Several cancer-related genes, including *BRCA1* and *MLL* (Myeloid/Lymphoid or Mixed-Lineage Leukemia; KMT2A, Lysine Methyltransferase 2A) contain many copies of the Alu element, which is the most abundant transposon in the human genome, accounting for about 10% its sequences [58,59]. The Alu element belongs to SINEs (short-interspersed nuclear elements), non-autonomous, non-coding transposable elements, a class of retrotransposons, which can play a role in cancer pathogenesis and be important for its therapy [60]. Therefore, when multiple DSBs occur in different chromosomes, they can be improperly recognized by SSA as its substrates based on the Alu repeats, flanking different DSBs. This may lead to pronounced chromosomal rearrangements, including chromosomal reciprocal translocations that may be essential in the pathogenesis of many cancers, especially in leukemias and lymphomas [61]. 

Chronic myeloid leukemia (CML) in adults is underlined by a reciprocal chromosomal translocation between chromosomes 9 and 22, t(9;22), resulting in the production of the *BCR/ABL1* fusion gene, whose expression gives the BCR/ABL1 protein, a constitutive tyrosine kinase activating cancer-related signaling pathways [62]. However, recent perspectives on CML development, especially in therapy-resistant cells, shows that the disease is driven by BCR/ABL1-independent pathways [63]. Nevertheless, BCR/ABL1 inhibition with imatinib (Gleevec), a tyrosine kinase inhibitor, resulting in ceasing disease progression, was the first success of cancer targeted therapy, changing CML from a fatal disorder into a chronic disease [64]. 

Salles et al., showed that BCR/ABL1 upregulated SSA and NHEJ but not HRR in human megakaryocytic and CML cell lines [65]. Kolomietz et al., showed that a considerable fraction of CML patients showed submicroscopic deletion of the 5′ region of *ABL* and the 3′ region of *BCR* genes, suggesting the involvement of SSA in the translocation event [66]. It was confirmed later by the Skorski’s lab with the use of a specific reported cassette integrated into genomic DNA [67]. This work also showed that not only BCR/ABL1, but also other fusion tyrosine kinases stimulated SSA activity. Later, we showed that BCR/ABL1 stimulated WRN mRNA and protein expression in part by c-MYC (MYC Proto-Oncogene, BHLH Transcription Factor)-mediated activation of transcription and Bcl-xL (BCL2 Like 1)–dependent inhibition of caspase-dependent cleavage, respectively [68]. It stimulated WRN-promoted SSA to protect BCR/ABL1-expressing cells against lethal effect of genotoxic stress.

SSA-induced chromosomal rearrangements might be attributed not only to SINEs, but also to LINEs (long-interspersed nuclear repeats), a class of non-LTR (long terminal repeat) retrotransposons, widely dispersed in the human genome, as showed by genome-wide analysis of LINE-LINE-mediated nonallelic homologous recombination [69]. 

Chromosomal translocations were observed with a high frequency when a DSB was located between two neighboring Alu repeats [70]. These translocations were largely (85%) attributed to repairing such DSBs by SSA. However, when DSB occurred between divergent Alu elements, the preferred repair pathway was shifted to NHEJ (93%). The essential SSA protein, RAD52, very effectively mediates annealing between identical Alu sequences, but shows a reduced efficacy with divergent Alu elements [71]. In summary, rearrangements resulting from the presence of divergent Alu repeats may result from MMEJ or a composite mechanism involving both MMEJ and SSA [34]. 

Hu et al., noted that when a DSB is close to one of two homologous repeats, then BIR, not SSA, is used to repair such damage [72]. This preference was stronger when the repeats were farther apart. The authors showed that SSA became ineffective in mammalian cells when the distance between a DSB and a repeat was increased to the 1-2 kb range, with BIR effectively acting over a long distance up to 200 kb, when DSB was located close to one repeat. That study may contribute to understanding mechanisms of oncogenesis associated with repeat-associated chromosomal translocation. Furthermore, the distance between two repeats is a major limiting factor for effective SSA.

## 5. SSA in BRCA-Deficient Cells and Cancer Cases

The mammalian *BRCA1* and *BRCA2* genes have their orthologs in lower organisms, which suggests their fundamental and likely tissue-specific functions [73]. BRCA deficiency (BRCAness) underlined by germline mutations in the *BRCA1/2* genes is typical for hereditary breast and ovarian cancers but also occurs in sporadic cases of these and other tumors [74]. The involvement of the *BRCA1/2* genes in cancer pathogenesis is assumed to be associated with their role in maintaining genomic stability in DNA repair, cell cycle regulation, and apoptosis [75]. The primary pathway with the involvement of BRCA1/2 is HRR, but BRCA1 may also influence SSA [76]. However, the involvement of BRCA1 in NHEJ is a matter of debate and it is hypothesized to promote a precise DNA repair in all cell-cycle phases through a direct modulation of DNA end-joining [77].

RAD52, the essential SSA protein, is an important target in cancer therapy [78]. Small molecule inhibitors were elaborated for RAD52 single-strand annealing activity, which was useful in synthetic lethality [79]. These compounds inhibited not only RAD52, but also the growth of BRCA1/2-deficient cells, typical for hereditary breast and ovarian cancers [80]. Therefore, RAD52 can be targeted in synthetic lethality in these two groups of cancers. However, mutations in the *BRCA1* gene are also present in sporadic breast cancer, including the most difficult cases to treat—triple-negative breast cancer [81]. Important results obtained in that study showed that inhibitors of RAD52 SSA activity did not interfere with the formation of the RAD51 foci in BRCA1-proficient cells. However, another work of the Skorski’s lab showed that RAD52 inhibitors attenuated residual HRR activity in BRCA1/2-deficient cells [82]. They showed that RAD52 inhibitors attenuated SSA in BRCA1/2-deficient cells and when acting in combination with PARP inhibitors, caused accumulation of DSBs and eradication of BRCA1/2-deficient, but not -proficient cells. They concluded that RAD52 inhibitors might increase the efficacy of PARP inhibitors-based treatment of cancers with BRCA-deficiency by the triggering dual synthetic lethality.

BRCA1/2 are reported to be downregulated in acute myeloid leukemia (AML) cell lines and primary cells from AML patients [83]. It was shown that RAD52 DNA aptamer inhibited proliferation, suppressed repair of etoposide-induced DSBs and promoted apoptosis in AML cell lines with downregulated BRCA1/2 [84]. These authors also observed S/G2 cell arrest affecting the STAT3 (Signal Transducer and Activator of Transcription 3) signaling pathway. Observed S/G2 arrest might result from an inhibition of the ATR/CHK1 complex. In summary, the authors suggested that RAD52 aptamers might be exploited in AML therapy in patients with a low expression of BRCA1/2.

Studies on the impact of variability of the RAD52 gene supplement the functional assessment of the role of RAD52 inhibition in killing BRCA-deficient cancer. Adamson et al., observed that the RAD52 p.S346X allele was associated with a reduced risk of breast cancer occurrence in carriers of pathogenic mutations in the *BRCA2* gene and to a lesser extent in their BRCA1 counterparts [85]. To investigate the mechanism underlying observed effects, the authors noted that the expression of RAD52 p.S346X suppressed the loss of DSBR in *RAD51^-/-^* mouse embryonic stem cells with the concomitant reduction in SSA frequency. Moreover, a reduction in the stimulation of SSA was observed upon depletion of *BRCA2*, suggesting a reciprocal role of RAD52 and BRCA1 in DSBR by SSA. The observed effect might be underlined by a decreased nuclear level of RAD52 p.S346X variant as compared with its wild-type counterpart. Altogether, defects in SSA underlined by RAD52 p.S346X variant might be associated with a reduced cancer risk in BRCA2-mutation carriers. 

The ring finger protein 168 (RNF168, RING Finger Nuclear Factor 168) plays an essential role in the recruiting factors involved in DSBR, including BRCA1 [86]. Munoz et al., showed that the depletion of RNF168 from the human osteosarcoma U2OS cell line resulted in SSA and HRR inhibition [87]. The authors postulated that RNF168 may be important in diagnosis of tumors with DSBR deficiency and this protein can be considered in therapy of BRCA1-deficinet tumors to prevent HDR restoration. Meghani et al.,, using material from BRCA1/2-deficient ovarian carcinoma patients, patient-derived lines, and an in vivo BRCA2-mutated mouse model, identified a microRNA, miR-493-5p, that induced platinum/PARPi resistance in BRCA2-mutated carcinomas [88]. This miRNA did not resume HRR because it was impaired by the lack of functional BRCA2, as was shown in another study [89]. Instead, it reduced the levels of factors involved in the genome maintenance. These included an increased stability of replication forks and decreased repair of DSB by SSA. The observed decrease in HRR and SSA induced by miR493-5p agreed with its impact on the DNA end resection proteins, including EXO1, MRE11, and BLM. Therefore, the authors hypothesized that miR-493-5p might increase PARPi and platinum resistance in BRCA2-deficient cells by diminishing the otherwise hyperactive SSA and promoting repair by NHEJ.

Obermeier et al., showed that the c.1592delT mutation in the *PALB2* (Partner and Localizer of BRCA2) increased SSA and B-NHEJ in a panel of lymphoblastoid cell lines obtained from heterozygous female carriers as compared with wild-type matched controls [90]. This mutation increases about six-fold the risk of breast cancer [91]. 

Ochs et al., showed that that silencing of the *53BP1* (*TP53BP1,* Tumor Protein p53 Binding Protein 1) gene or limiting the capacity of its product to bind damaged chromatin resulted in a hyper-resection of DSBs ends and switch from error-free gene conversion by RAD51 to SSA by RAD52 [92]. These results shed some light on causes and consequences of synthetic viability resulting from TP53BP1 silencing in cells with deficient BRCA1—such cells survive despite DSBs overproduction due to RAD52-mediated SSA, which may accelerate genome instability. Therefore, BRCA1- and TP53BP1-deficient cancer cells may be eliminated by SSA/RAD52 inhibition.

Hanamshet and Mazin showed that N-termina domain (NTD) of RAD52 formed nuclear foci upon DNA damage in BRCA-deficient human cells and promoted DNA repair by HRR and SSA [93]. Moreover, these authors showed that mutations in NTD failed to maintain the viability of BRCA-deficient cells.

To correlate functions of HRR and SSA with missense mutations in *BRCA1,* Towler et al., evaluated the functions of 29 *BRCA1* mutations [74]. They found that all pathogenic variants were defective for HRR and all their non-pathogenic counterparts were fully HRR functional. On the other hand, SSA was not only defective in all pathogenic variants, but in some non-pathogenic mutants as well. 

In general, the SSA repair pathway distinguishes BRCA1- and BRCA2-deficient cells. The former exhibit diminished HRR and SSA, while the latter display decreased HHR but a hyperactive SSA, which may be responsible for chromosomal aberrations in these cells [94]. 

## 6. SSA in Cancer Clinic

In a case-control study with 38 women with hereditary ovarian cancer, 40 women with sporadic ovarian cancer, and 34 healthy controls, Deniz et al., showed that SSA, not NHEJ, compensated for compromised HRR due to BRCA defects [95]. These authors observed increased SSA in lymphocytes of blood derived from women with ovarian cancer and familial risk of this disease. Lymphocytes from these patients were more sensitive to carboplatin than the cells obtained from controls. Therefore, SSA may play a role in the pathogenesis of ovarian cancer and a relationship between administration of platinum-based drugs and SSA may be exploited in ovarian cancer therapy. Of note, these authors observed that NHEJ, both canonical and alternative, did not discriminate between ovarian cancer patients and healthy individuals. 

Kohzaki et al., observed that tumorigenic cell lines with the deletion in the C-terminal region (ΔC) of the *RECQL4* (RecQ Like Helicase 4) gene, displayed hypersensitivity to ionizing radiation and cisplatin [70]. Both kinds of anticancer treatment resulted in an increased number of RAD52/RPA2 foci. HCT116 colon cancer cells with such a deletion showed increased SSA and decreased MMEJ, suggesting that RECQL4 may be an important regulator of the choice of DSBR. From the clinical point of view, cancer patients with mutations in the *RECQL4* gene and scheduled for DSB-induced anticancer treatment may be given chemicals to modify SSA to increase the efficacy of the treatment.

It was shown that FANCA, a component of the Fanconi anemia (FA) core complex, catalyzed bidirectional SSA and strand exchange (SE) at a level similar to RAD52 [96]. However, a disease-causing mutant of FANCA did not do so. Moreover, FANCG directly stimulated FANCA-dependent SSA activity. It was concluded that FANCA played a direct role in SSA through the catalysis of SE independently of the canonical FA pathway and RAD52. In their subsequent work, Liu et al., showed that a steroid lactone withaferin A (WA) at a submicromolar concentration inhibited SSA by downregulation of FANCA [97]. WA action also resulted in increased sensitivity to mitomycin C (MMC) and accumulation of DSBs in human bone osteosarcoma epithelial cells. MMC induced DNA interstrand crosslinks that challenge the cell as there is not any uniform DNA repair system dedicated to repair such kind of DNA damage and DSBs are intermediate of ICL repair [98]. Therefore, WA may be considered an agent to increase the efficacy of DSB- or ICL-inducing cancer therapy.

Bortezomib is a first-in-class proteasome inhibitor targeting the 26S proteasome, a multi-subunit complex of the ubiquitin-proteasome system [99]. Bortezomib interfere with the FANC signaling, so it may influence DSBR [100]. Howard et al., showed that bortezomib inhibited end resection and reduced HRR, B-NHEJ and SSA in U2OS cells [37].

Keimling et al., found in a case-control study an increase in SSA and B-NHEJ in peripheral blood lymphocytes (PBLs) of women with defined familial history of breast and/or ovarian cancer as compared with women with no previous cancer or family history of breast cancer (controls) [101]. The same relationship was observed in breast cancer patients as compared to controls. Furthermore, increased SSA in cancer patients was associated with young age (<50 years) at diagnosis. These findings suggest that cancer-related SSA alterations can be detected in PBLs so they may be exploited in diagnosis and therapy of DSBR-defective tumors.

## 7. SSA in Therapeutic Genome Editing

Genome editing has created an ample opportunity to plan therapeutic strategies in many human diseases, including cancer. The 2020 Nobel-prized CRISPR (Clustered Regularly Interspaced Short Palindromic Repeats)/Cas9 (CRISPR Associated 9) is currently the most common system to edit the human genome and genomes of other organisms. It is considered to have a high potential in current and prospective cancer diagnosis and therapy [102,103]. CRISPR/Cas9 was applied first in phase I clinical cancer in non-small cell lung cancer [104]. Furthermore, it is successfully used to explore mechanisms of cancer pathogenesis through modification of cancer cell lines and creation of animal models of human cancers [105]. 

Despite many spectacular and unquestionable successes of CRISPR/Cas9 in cancer biology and medicine, it still presents some serious challenges. The off-target effects of this system, resulted from insertion and deletions at non-targeted regions in the genome, make it difficult to target a specific genomic locus [106]. Therefore, efforts are undertaken to improve the CRIPR/Cas technology and reduce any undesired off-target effects. The combination of CRISPR/Cas with Cre/LoxP or the *piggyBac* transposon increases the efficacy of the technology, but still leaves behind traces (“scars”) in the edited genome [107,108].

To increase precision and efficacy of genome editing, Li et al., combined the SSA-mediated repair mechanism with CRISPR/Cas [109]. They developed an integrating cassette with positive/negative selection markers flanked by direct repeats with specific mutations in target sites. This cassette was integrated in a target sequence after CRISPR/Cas9-mediated homologous recombination. After positive clone selection, the cassette was removed by SSA. This method employed at *CCR5* (C-C Motif Chemokine Receptor 5) and *APP* (Amyloid Beta Precursor Protein) loci resulted in precise genome editing efficiency up to 45.83 and 68%, respectively.

Phenotypical distinguishing of cells with genomes edited by CRISPR/Cas9 is one of the most serious challenges of this technology. Ren et al., addressed this problem by the development of two dual-reporter surrogate systems based on NHEJ and SSA – NHEJ-RPG and SSA-RPG [110]. They used CRISPR/Cas9 and the zinc finger nucleases (ZFNs) to induce DSB and they found that the efficacy and sensitivity of the SSA-RPG reporter with direct repeat of length more than 200 bp were much higher than that of the NHEJ-RPG reporter. The SSA-RPG reporter caused the enrichment for indels in several loci with 6.3- to 34.8-fold of non-selected cells. These systems presented alternative enrichment choices either by puromycin selection or fluorescence-activated cell sorting.

In a similar study, Jinqing et al., observed that the use of a SSA reporter system improved the activity of genome editing by CRISPR/Cas9 by 5 folds and even more when ZFN technology was employed to edit a porcine genome [111].

In that context it is worth noting, that SSA plays a major role in DSBR after CRISPR/Cas9 cleavage in the protozoan *Leshmania,* which is deficient for C-NHEJ and preferentially employs MMEJ and occasionally HRR to repair DSB [112]. These studies showed the action of SSA when repeats were 77 kbp apart, which resulted in the deletion of 15 genes. Although not directly associated with cancer this study shows a role SSA may play in DSBR after therapeutic use of CRISPR/Cas9 technology.

In summary, SSA can be used to refine different genome editing technologies, including CRISPR/Cas9. SSA can be directly applied along with a basic editing technique to reduce its off-target effects and enrich the population of edited cells or can be exploited to improve the technique itself by providing information on requirements of its effective action, including the length of template for HRR or the topology of repeats for NHEJ.

## 8. Conclusions and Perspectives

At present the physiological role of SSA is unclear [113]. In general, HRR and NHEJ seem to be sufficient to repair most DSBs in every cellular context. However, the question about the evolutionary role of SSA is not easy to answer. It is important that in some cases analysis of repair outcome does not allow a definite conclusion to be drawn as to which repair pathway was involved in an outcome. This especially concerns discrimination between crossover and non-crossover products. The latter may result from HRR, B-NHEJ, or SSA. Moreover, even assays that are apparently dedicated to NHEJ do not distinguish between B-NHEJ and SSA. This is further complicated by the involvement of RAD52 in all three main DSB repair pathways [114]. Therefore, certain reports indicating the involvement of B-NHEJ in some effects may actually be related to SSA alone or acting in conjunction with another DSBR pathway. 

In general, SSA seems to be a somehow “forgotten” or underestimated DSBR pathway in research, as several reports on synthetic lethality consider only HRR and NHEJ, including B-NHEJ, as DSBR pathways, and mention SSA as a likely operating system [115,116]. It is especially important in the context of the involvement of RAD52 in B-NHEJ and HRR [117].

Homology-directed repair has at least two general aspects in cancer transformation. It prevents or ameliorates genomic instability, but on the other hand it may result in loss of heterozygosity (LOH), important in cancer initiation [118]. Such an effect is due to replacing of a damaged allele of a gene, with its undamaged counterpart carrying a cancer-related mutation. This may result from the action of HRR, SDSA, and BIR, but the involvement of SSA in LOH is rather hypothetical [119]. A more likely scenario is associated with gene dosage, when a copy of a gene is inactivated due to its fragment deletion [120].

The involvement of MMR factors in SSA regulation or mutual interconnection between MMR and NHEJ is still an open issue, especially since MMR has both pro- and anti-recombinational properties [121]. 

Apart from repair, RAD52 is involved in many aspects of DNA and RNA metabolism, including replication, which are exploited by viruses to facilitate their propagation [117,122]. This naturally makes RAD52 a target in antiviral therapy and as some cancers are virus-dependent, RAD52 and SSA may be especially important in such cancers. Therefore, small molecular-weight modulators of RAD52 activity may be considered as drugs in the therapy of virus-related cancers. It seems that RAD51 may be critical for the role of SSA in cancer, although it is not directly related to the DSBR pathway. Stark et al., showed that RAD51 limited mutagenic potential of HDR—in the most extreme case they found that disruption of RAD51 resulted in a more than 90-fold shift in DSBR pathway usage toward SSA [94].

HDR may be directed by an RNA template, which has been discovered and mechanistically explored in yeast [123]. A recent study showed that RNA-templated recombination in postmitotic neurons occurred with the involvement of RAD52 and that the affinity of RAD52 to RNA might exceed that to DNA [124].

In conclusion, SSA may contribute to cancer transformation even in its normal functioning as it may increase genomic instability and induce chromosomal translocation. Its inhibition may also contribute to the death of cancer cells deficient in other DSBR pathways, especially HRR with impaired RAD51 or BRCA1/2. This makes SSA a valuable tool in synthetic lethality-based cancer therapy. SSA may be also exploited in therapeutic genome editing to increase its efficacy and reduce off-target effects. Further studies on the regulation of SSA, especially in the context of global DSBR, are needed to determine SSA’s role in cancer transformation and its potential in cancer therapy.

## Figures and Tables

**Figure 1 ijms-22-02167-f001:**
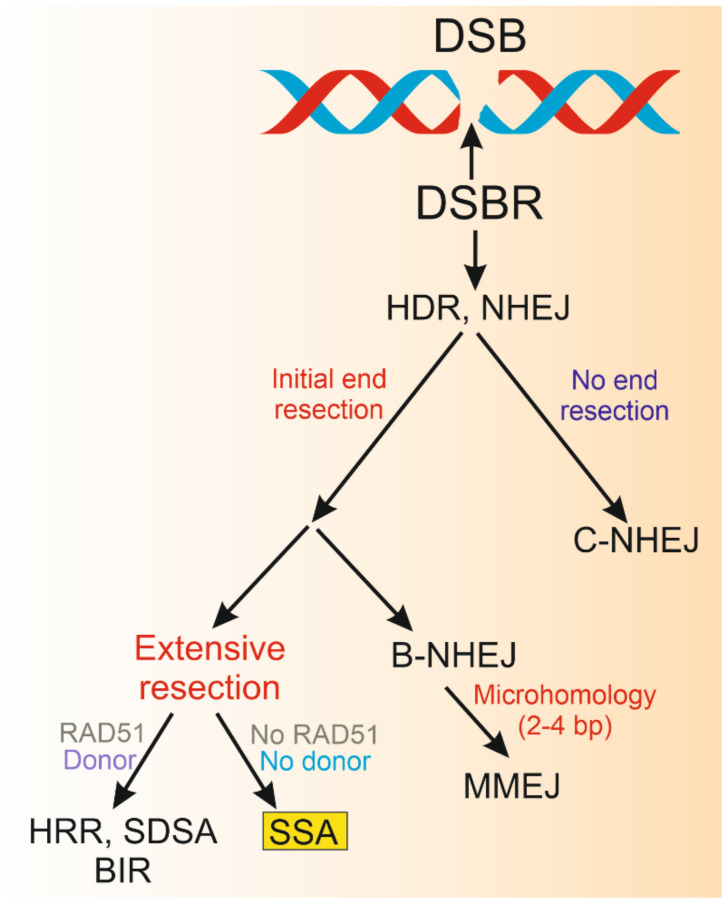
Single-strand annealing (SSA) is a DNA double-strand break (DSB) repair (DSBR) system which uses homologous repeats flanking a DSB. Therefore, SSA is a pathway in the homology-directed repair (HDR), which includes homologous recombination repair (HRR), synthesis-dependent strand annealing (SDSA), and break-induced recombination (BIR). The other group of DSBR pathways, non-homologous end joining (NHEJ), in its canonical form (C-NHEJ) is independent of homology, but its backup variant (B-NHEJ) uses microhomology (2-4 bp), taking the form of microhomology-mediated end joining (MMEJ). HDR and MMEJ, in contrary to C-NHEJ, require DNA end resection to expose 3′ single-stranded DNA fragments. These fragments may undergo further, extensive resection and invade double-stranded DNA (Donor), which provides a template DNA as in HRR, SDSA, and BIR when active RAD51 (RAD51 Double Strand Break Repair Protein) is present. When there is no active RAD51 or, despite its presence, no donor DNA, 3′ ss overhangs produced in the end resection process may be a substrate for SSA. After initial resection B-NHEJ may occur, especially when short homologous sequences are located near the DSB (MMEJ).

**Figure 2 ijms-22-02167-f002:**
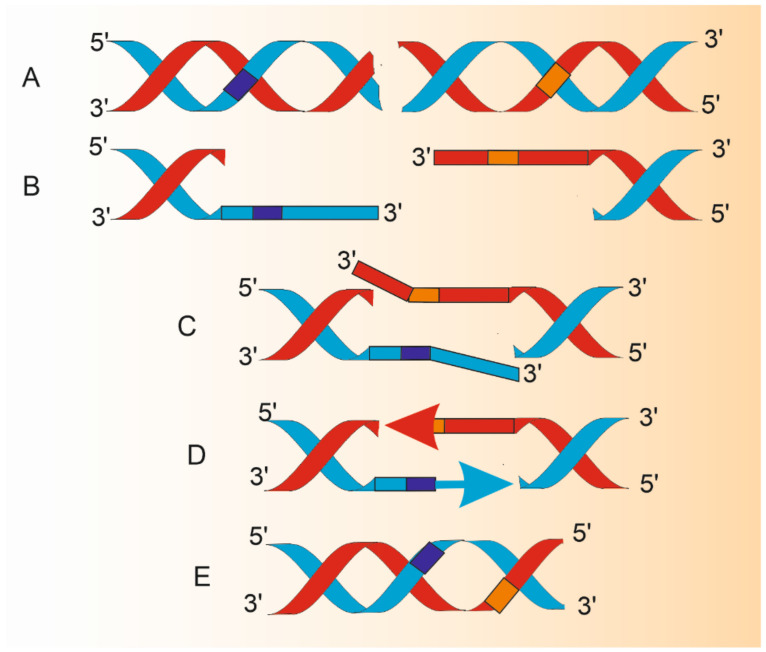
Changes in DNA structure during single-strand annealing (SSA). Double-strand break (DSB) in DNA having homologous repeats flanking DSB (dark blue and orange boxes) is a substrate for SSA (**A**). A combined action of DNA helicases and nucleases results in 5′ ends resection and production of 3′ overhangs (**B**) that are annealed to juxtapose the repeats (**C**). Protruding 3′ single-stranded fragments are removed by endonucleolytic cleavage and remaining 3′ ends are extended by a DNA polymerase (arrows) to synthesize DNA fragments with the use of an intact template of complimentary strand (**D**). DNA ligase reseals, lacking phosphodiester bonds. Resulting DNA is shorter than the parental molecule by the distance between homologous repeats and the length of one repeat (**E**). Therefore, SSA is potentially mutagenic even in its normal functioning. The proteins involved in A-E are described in the main text. This figure does not correspond to actual proportions within DNA processed by SSA.

**Figure 3 ijms-22-02167-f003:**
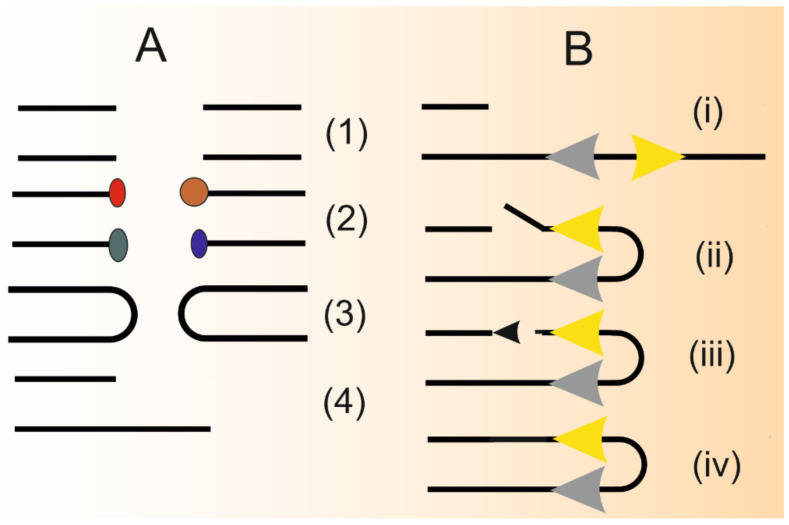
Different structures of DNA double-strand breaks (DSBs) (**A**): “clean” termini (3′-OH or 5-P) (1), termini blocked by different (color and shape) chemical groups (2), hairpin-structured ends (3) and one-ended DSB (4). Single-strand annealing (SSA) in one-ended DSB with inverted homologous repeats (**B**). Grey and yellow arrows depict homologous, but not necessarily identical sequence oriented in opposite directions (i) that are annealed and protruding non-homologous 3′ ends are removed by an endonucleolytic cleavage (ii); remaining gap is fulfilled by a DNA polymerase (black arrow) (iii) and lacking phosphodiester bond is sealed by DNA ligase (iv).

**Figure 4 ijms-22-02167-f004:**
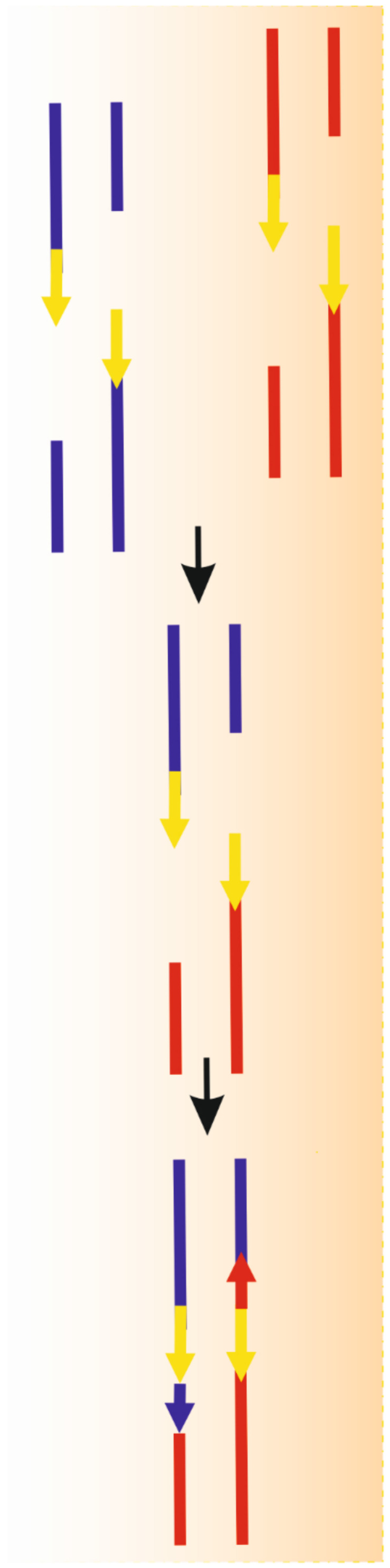
Single-strand annealing (SSA) may induce chromosomal translocations. Different (heterologous) chromosomes may experience double-strand breaks (DSBs) that may be flanked by the same homologous repeats (yellow arrows). SSA machinery may combine repeats from different chromosomes and produce a fusion chromosome, in which gaps are filled by a DNA polymerase (red and blue arrows) and DNA ligase seals the lacking phosphodiester bond. Protruding non-homologous single-stranded fragments resulting from the annealing process are not presented.

## Data Availability

Data sharing not applicable.

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
