# Peer review of "Single-Strand Annealing in Cancer"

_ijms, 2021, doi:10.3390/ijms22042167_

Round 1

Reviewer 1 Report

Janusz Blasiak contributes an excellent review about SSA in the context of DSB repair, cancer,  synthetic lethality and genome editing. 

The review is up-to-date and addresses different critical issues such as 

- current perception of SSA in DSB repair

- involvement in formation of chromosomal rearrangements in cancer

- synthetic lethality (e.g., RAD52 inhibition in distinct cancers)

- genome editing

It raises several interesting questions, e.g.,

- Is there a sharp boundary separating homology lengths specific for these systems? (SSA vs MMEJ)

- potential clinical exploitation by synthetic lethality

There are few syntax errors that should be corrected, e.g.,

As DBSs are a direct threat...

Both SSA and MMEJ anneal homologous sentence of DNA...

...human megakaryotic and CML cell lines

...failed to maintain the viqability of BRCA-deficient cells.

...prevents or ameliorate gnomic instability...

Author Response

Janusz Blasiak contributes an excellent review about SSA in the context of DSB repair, cancer,  synthetic lethality and genome editing.

The review is up-to-date and addresses different critical issues such as

- current perception of SSA in DSB repair

- involvement in formation of chromosomal rearrangements in cancer

- synthetic lethality (e.g., RAD52 inhibition in distinct cancers)

- genome editing

It raises several interesting questions, e.g.,

- Is there a sharp boundary separating homology lengths specific for these systems? (SSA vs MMEJ)

- potential clinical exploitation by synthetic lethality

Comment: There are few syntax errors that should be corrected, e.g.,

As DBSs are a direct threat...

Both SSA and MMEJ anneal homologous sentence of DNA...

...human megakaryotic and CML cell lines

...failed to maintain the viqability of BRCA-deficient cells.

...prevents or ameliorate gnomic instability...

Answer: I have corrected all these errors.

Reviewer 2 Report

This review depicts the role of SSA in DSB repair and cancer. While providing extensive information on this field, it lacks mechanistic insights, more particularly (but not limited to) on SSA pathway, SSA regulation (especially in regard to homologous recombination (HR)) and RAD52 functions. Therefore, some informations are not sufficiently explained or even misleading.

SSA is widely considered as an error-prone backup pathway, but has also been proposed to represent an error- free repair system in a specific context: at stalled forks by suppressing tandem duplications at sites of aberrant replication fork restart (Scully et al. 2019, Nat Rev Mol Cell Biol). However, this is still hypothetical and needs experimental validation. Thus, SSA cannot be presented as a major cellular DSB repair pathway at the same level as HR and classical NHEJ, as already done in the abstract and latter in the text.

Author definition of HRR (page 3) is confusing, and somewhat erroneous. HRR is presented as HR that processes double Holliday junctions and that differs from other HR pathways, including meiotic HR. However, meiotic HR is precisely dedicated to double Holliday junction processing between homologous chromosomes, with specific factors (not only DSB induction factors as stated by author) including the specialized recombinase DMC1, ZMM proteins and specialized resolvase (thus DNA repair proteins). Besides, author refers to Spo11 (page 2) as inducing DSBs in a variety of processes. Spo11 actually induces DSBs specifically in meiosis, thus could author explain more clearly in which other processes it could be involved? In contrast, somatic cells mainly use SDSA and not Holliday junction resolution.

Figure 1 must be removed as it does not give any relevant information and is misleading: MMEJ is a (micro)homology directed repair).

End resection is a crucial mechanism to consider as its initiation regulates DSB repair pathway choice between cNHEJ and others, and longer resection regulates between MMEJ and HR/SSA. Then, RAD51 forms a nucleofilament with the 3'-end ssDNA and initiates strand invasion of the homologous dsDNA template. Failure to form RAD51 nucleofilament (like in BRCA mutants) favors extensive resection and thus SSA. These are essential messages that are not clearly explained in this manuscript. At least a figure should be included, and a dedicated paragraph. Instead, author gives scattered informations, some of which are wrong:

  • page 5: the role of MRN and CtIP is not clear.
  • page 6: "Several other factors may be included in the resection reaction, including MRN and BRCA2 (reviewed in [22])." The role of MRN as already been exposed (page 5); and BRCA2 is not involved in resection, but rather BRCA1.
  • page 6: RAD51 and RPA do not compete for ssDAN binding. RPA firstly binds ssDNA and is then displaced through the action of mediators (BRCA2, PALB2) to allow RAD51 filament formation.
  • page 6: strand invasion must be explained
  • page 11: what is the difference between the "microSSA" type of NHEJ and MMEJ?
  • page 12: To better understand the role of RAD52 in HR, author refers to S. cerevisiae. However, author may know that yeast Rad52 as a distinctive role of mediator (displacement of RPA/SSB) that in mammals is accomplished by BRCA2 and PALB2.
  • page 17: author claims that RAD52 is involved in the main repair pathways including NHEJ. However, NHEJ suggests classical NHEJ, in which RAD52 has no role. RAD52 may promote microhomology directed repair (B-NHEJ).

Section 3: SSA in genomic instability and DNA repair defects in cancer (pages 7 to 9). There are only two lines considering DNA repair defects (RAD51 suppression activate SSA)!. This requires an independant section including BRCA1/2 (pages 12 and 13), RNF168, SIRT1, miR493-5p, PALB2 etc...

Author claims that RAD52 as a role in killing BRCA-deficient cels (page 13), while RAD52-dependent SSA actually rescue BRCA-dependent HR defects. If author refers to RAD52 aptamer (page 12), these are DNA sequences that bind and inhibits RAD52. Their use in BRCA-defective cells inhibits SSA and thus kill cancer cells.

RNF168 page 13 : why restoring HR in BRCA1 deficient tumor would improve cancer therapy?

Diallyl disulfide (DDS, page 14) inhibits resection and thus HR repair, not only SSA. This information is not relevant here.

Conclusion page 17. SSA do not contribute to HR-deficient cell death. On the contrary, SSA inhibition induce synthetic lethality in these genetic background.

The manuscript requires extensive proofreading.

Author Response

Comment: This review depicts the role of SSA in DSB repair and cancer. While providing extensive information on this field, it lacks mechanistic insights, more particularly (but not limited to) on SSA pathway, SSA regulation (especially in regard to homologous recombination (HR)) and RAD52 functions. Therefore, some informations are not sufficiently explained or even misleading.

Answer: Thank you very much for your essential, in-depth comments. In general, this review focuses on the role of SSA in cancer. The mechanism underlying SSA action is not fully known and several hypotheses, more or less justified, have been addressed. I think that this review should not contain more text about possible SSA mechanisms than its role in cancer, although often specific mechanism underlining this role is not (fully) known. Surely, I have tried to correct misleading information and supplement lacking details according to the Reviewer’s suggestions.

Comment: SSA is widely considered as an error-prone backup pathway, but has also been proposed to represent an error- free repair system in a specific context: at stalled forks by suppressing tandem duplications at sites of aberrant replication fork restart (Scully et al. 2019, Nat Rev Mol Cell Biol). However, this is still hypothetical and needs experimental validation. Thus, SSA cannot be presented as a major cellular DSB repair pathway at the same level as HR and classical NHEJ, as already done in the abstract and latter in the text.

Answer: Surely, SSA should not be perceived as a major DSBR pathway, in contrary to HR and NHEJ, and all I wanted to say, was that it was an autonomous DSBR system, regarding the proteins and biochemical reactions that are involved, and not directly related to HR or NHEJ. Some publications, including Friedberg et al. “DNA Repair and Mutagenesis”, present SSA as a just the third DSBR pathway, besides HR and NHEJ.

To address this comment, I have made the following changes:

  1. I have replaced the fragment:

“Three major cellular pathways to repair DSBs are homologous recombination repair (HRR), non-homologous end joining (NHEJ) and single-strand annealing (SSA). SSA uses homologous repeats flanking a DSB to join DNA ends and has a mutagenic potential as it removes DNA fragments between repeats.”

with the following fragment:

“In humans, DSBs are repaired mainly by non-homologous end joining (NHEJ) and homologous recombination repair (HRR). Single-strand annealing (SSA), another DSB repair system, uses homologous repeats flanking a DSB to join DNA ends and is error-prone as it removes DNA fragments between repeats along with one repeat.”

  1. I have changed the sentence:

“In a more precise categorization, three main pathways of DSBR can be considered: homologous recombination repair (HRR), non-homologous end joining (NHEJ) and single-strand annealing (SSA) (reviewed in [8]).”

into:

“Humans and other higher organisms repair DSBs primarily by non-homologous end joining (NHEJ) and homologous recombination repair (HRR). Another DSBR pathway is single-strand annealing (SSA), but it seems to be of a minor significance as compared with NHEJ and HRR (reviewed in [8]).”

I have changed the fragment:

“Single-strand annealing is sometimes suggested to be directly related to its counterparts in DSBR – HRR and NHEJ, as in general it uses homology similarly to HRR, and when this homology is short, it resembles microhomology-mediated NHEJ [12]. However, due to genetic components and biochemical reactions in SSA, it should not be considered as an HRR or NHEJ variant.”

into:

“Despite of some mechanistic similarities, SSA should not be considered as an HRR or NHEJ variant as its functioning is underlined by different proteins and biochemical reactions [12].”

I have added the following information about hypothetic error-free action of SSA:

“Scully et al. considered a state, when two DNA ends of a tandem duplication were repaired by SSA so the duplicated segments overlapped each other changing the duplication into original single copy region [8]. In that circumstances, SSA would accomplish a conservative at a stalled replication fork by preventing the formation of tandem repeats by abnormally restarted replication fork. Therefore, error-prone SSA may hypothetically turn into error-free repair at stalled replication fork.”

Comment: Author definition of HRR (page 3) is confusing, and somewhat erroneous. HRR is presented as HR that processes double Holliday junctions and that differs from other HR pathways, including meiotic HR. However, meiotic HR is precisely dedicated to double Holliday junction processing between homologous chromosomes, with specific factors (not only DSB induction factors as stated by author) including the specialized recombinase DMC1, ZMM proteins and specialized resolvase (thus DNA repair proteins). Besides, author refers to Spo11 (page 2) as inducing DSBs in a variety of processes. Spo11 actually induces DSBs specifically in meiosis, thus could author explain more clearly in which other processes it could be involved? In contrast, somatic cells mainly use SDSA and not Holliday junction resolution.

Answer: Absolutely! I have changed the sentence (page 2):

“Programmed DSBs can be induced by the Spo11 (SPO11 Initiator of Meiotic Double Stranded Breaks) endonuclease in a variety of processes [5].”

into:

“Programmed DSBs are induced by the Spo11 (SPO11 Initiator of Meiotic Double Stranded Breaks) endonuclease in meiosis [5].”

I have also removed the sentence (page 3):

“Furthermore, HRR is often presented as HR, which is somehow elusive, as HR may be realized in several pathways, including meiotic HR with proteins that are not involved in DNA repair, but rather DNA damage induction.”

I think that with these changes, the HRR definition I presented can be accepted. However, if I knew other terminology corresponding with the classification shown in Figure 1 (formerly Figure 2), I would follow it.

Comment: Figure 1 must be removed as it does not give any relevant information and is misleading: MMEJ is a (micro)homology directed repair).

Answer: I have removed it along with the preceding fragment:

“The classification presented in Figure 1 is only a proposition for the need of this review, as it may not be in line with several other papers. For instance, HDR is frequently understood as HRR, which excludes SSA from HDR repair.”

Comment: End resection is a crucial mechanism to consider as its initiation regulates DSB repair pathway choice between cNHEJ and others, and longer resection regulates between MMEJ and HR/SSA. Then, RAD51 forms a nucleofilament with the 3'-end ssDNA and initiates strand invasion of the homologous dsDNA template. Failure to form RAD51 nucleofilament (like in BRCA mutants) favors extensive resection and thus SSA. These are essential messages that are not clearly explained in this manuscript. At least a figure should be included, and a dedicated paragraph. Instead, author gives scattered informations, some of which are wrong.

Answer: I have supplemented Figure 2 (now Figure 1) with RAD51, initial resection, extensive resection, invasion and search for homology. I have also added the following paragraph:

“As presented in Figure 1, end resection is a crucial mechanism in the choice between C-NHEJ and HDR. However, HDR includes several pathways, thus the question is whether it may be involved in discrimination between HRR/SDSA/BIR and SSA/MMEJ. Initial, relatively short resection is a signal for RAD51 (RAD51 Double Strand Break Repair Protein) to form a nucleofilament with ssDNA overhang resulting from the resection of the complimentary strand Moreover, HRR, SDSA and BIR are initiated by strand invasion, which is led by RAD51 forming a nucleofilament with ssDNA overhang resulting from the resection of the complimentary strand. Neither SSA, nor MMEJ requires strand invasion. However, strand invasion may be effective only when a donor DNA is juxtaposed, but the presence of donor DNA is not enough when there is no functional RAD51 as is in the case of BRCA mutants (BRCAness) [8]. In general, ablation of RAD51 stimulates SSA and MMEJ, but does not activate C-NHEJ. In such case, an extensive resection is favored, creating a substrate for SSA and MMEJ (Figure  X). It is possible that If the nearby repeats to DSB are short (4-5 bp), MMEJ is preferred, when they are longer – SSA may be activated. Moreover, the distance between DSB and nearby repeats may be important. Moreover, it was shown that repressing SSA and MMEJ by RAD51 resulted from its non-catalytic binding to ssDNA and was independent of its invasion/strand exchange catalytic activity [1].”

Comment: page 5: the role of MRN and CtIP is not clear.

Answer:  I have changed the paragraph:

“At the initial stage, DNA ends are resected by a complex interaction between the structure-specific nucleases MRE11 (MRE11 Homolog, Double Strand Break Repair Nuclease) and CtIP (RB Binding Protein 8, Endonuclease) [17,18] (Figure 3). MRE11 is a component of the MRN (MRE11, RAD50 (RAD50 Double Strand Break Repair Protein) and NBS1 (Nibrin)) complex and its two remaining components are also involved in the resection and this is RAD50 that facilitates the formation of bridges between DNA ends and NBS1 recruits ATM and CtIP to DSBs [19,20]. Finally, MRE11 uses its nuclease activity to resect the DNA ends [21]. Such resected DNA ends ranging about 100 nt may be a substrate for MMEJ, a variant of B-NHEJ [22]. Longer single-strand (ss) stretches (extensive resection) are generated by the combined action of helicases and endonucleases, including DNA2 (DNA Replication Helicase/Nuclease 2), BLM (BLM RecQ like helicase), WRN (WRN RecQ Like Helicase), CtIP and EXO1 (Exonuclease 1) [23]. Such long 3’-ss stretches may be a substrate for HRR or SSA [24]. The choice between these systems is not completely clear and many factors can be involved.”

into:

“Although not all details of DSBR choice are fully known, the resection step is it major decisive process. At the initial stage, DNA ends are resected by the interaction between the structure-specific nucleases MRE11 (MRE11 Homolog, Double Strand Break Repair Nuclease) and CtIP (RB Binding Protein 8, Endonuclease) [17,18]. MRE11 is a component of the MRN (MRE11, RAD50 (RAD50 Double Strand Break Repair Protein) and NBS1 (Nibrin)) complex, which binds DNA through the association of MRE11 and NBS1 with the A and B Walker motifs in RAD50 and the interaction with the extended coiled-coil tail of RAD50 [19]. These interactions allow the MRN complex to form a bridge between DNA ends. CtIP and ATM dimers are then recruited to the DSB site by specific domains in NBS1. CtIP functions together with MRN, but its nuclease activity is likely required to process complex DNA ends to enable the action of end-resecting exonucleases that need 3-OH and 5’-P ends [20]. As MRE11 has 3’-5’ exonucleolytic activity, opposite to 5’-3’ counterpart required for 3’-overhangs formation in end resection, it firstly induces a nick in DNA using its endonucleolytic activity and then degrades DNA towards DSB producing 3’ ssDNA tails [21]. Such resected DNA ends ranging about 100 nt may be a substrate for MMEJ, a variant of B-NHEJ [22]. Longer single-strand stretches (extensive resection) are generated by the combined action of helicases and endonucleases, including DNA2 (DNA Replication Helicase/Nuclease 2), BLM (BLM RecQ like helicase), WRN (WRN RecQ Like Helicase), CtIP and EXO1 (Exonuclease 1) [23, 24].”

Comment: page 6: "Several other factors may be included in the resection reaction, including MRN and BRCA2 (reviewed in [22])." The role of MRN as already been exposed (page 5); and BRCA2 is not involved in resection, but rather BRCA1.

Answer: I have removed that sentence. The involvement of BRCA1 in the resection reaction was presented in the fragment preceding that sentence.

Comment: page 6: RAD51 and RPA do not compete for ssDAN binding. RPA firstly binds ssDNA and is then displaced through the action of mediators (BRCA2, PALB2) to allow RAD51 filament formation.

Answer: I have replaced the fragment:

“Resected long 3’-ss ends are competitively targeted by the RAD51 (RAD51 Double Strand Break Repair Protein) and RPA (Replication Protein A) complex [31]. The binding of RPA to 3-ss ends prevents annealing between short homologies, avoids MMEJ and supports SSA, which is independent of RAD51 as it does not require a donor sequence and strand invasion [32].”

with:

“Resected long 3’-ss ends are targeted by RPA (Replication Protein A) and RAD51 [31]. RPA firstly binds ssDNA and is then displaced through the action of mediators, including BRCA2 and PALB2 (Partner and Localizer of BRCA2) to allow RAD51 filament formation. Therefore, in the absence of functional RAD51, as is the case in BRCA mutants, RPA binding to 3-ss ends prevents annealing between short homologies, avoids MMEJ and supports SSA, which is independent of RAD51 as it requires neither donor sequence nor strand invasion [32].”

Comment: page 6: strand invasion must be explained

Answer: I have added the following fragment:

“RAD51 leads invasion of the damaged duplex on its undamaged, homologous counterpart. This is facilitated by the two distinct sites in the RAD51 filament: primary, accommodating ssDNA and secondary, which can accommodate double dsDNA in a weak, transient and independent of DNA sequence. In this way, RAD51 may check long stretches DNA searching for homology. When RAD51 finds it, primary and secondary sites exchange ssDNA, forming new duplex and promoting a reciprocal invasion of the template on the damaged DNA.”                                                                          

Comment: page 11: what is the difference between the "microSSA" type of NHEJ and MMEJ?

Answer: No difference, just a historical note how MMEJ was perceived. I have removed the sentence:

“In addition, Elliot et al. considered a “micro-SSA” type of NHEJ with only a few bp identical sequences [68].”

Comment: page 12: To better understand the role of RAD52 in HR, author refers to S. cerevisiae. However, author may know that yeast Rad52 as a distinctive role of mediator (displacement of RPA/SSB) that in mammals is accomplished by BRCA2 and PALB2.

Answer: I am sorry, I do not understand this comment in the context of the manuscript and therefore I have not undertaken any action to address it.

Comment: page 17: author claims that RAD52 is involved in the main repair pathways including NHEJ. However, NHEJ suggests classical NHEJ, in which RAD52 has no role. RAD52 may promote microhomology directed repair (B-NHEJ).

Answer: I have changed NHEJ into B-NHEJ in the following sentences:

“The latter may result from HRR, NHEJ or SSA. Moreover, even assays that are apparently dedicated to NHEJ do not distinguish between NHEJ and SSA.”

Comment: Section 3: SSA in genomic instability and DNA repair defects in cancer (pages 7 to 9). There are only two lines considering DNA repair defects (RAD51 suppression activate SSA)!. This requires an independant section including BRCA1/2 (pages 12 and 13), RNF168, SIRT1, miR493-5p, PALB2 etc...

Answer: I have transferred the paragraph on SIRT1 from Section 5 into Section 3 and added the following information at the end of that section:

“DNA repair defects underlined by BRCA1/2 deficiency and chromosomal translocations will be considered in the subsequent sections.”

Comment: Author claims that RAD52 as a role in killing BRCA-deficient cels (page 13), while RAD52-dependent SSA actually rescue BRCA-dependent HR defects. If author refers to RAD52 aptamer (page 12), these are DNA sequences that bind and inhibits RAD52. Their use in BRCA-defective cells inhibits SSA and thus kill cancer cells.

Answer: I have changed the sentence:

“In line with the functional assessment of the role of RAD52 in killing BRCA-deficient cancer cells are studies on the impact of genetic variability of RAD52 on this effect.”

into:

“Studies on the impact of variability of the RAD52 gene supplement the functional assessment of the role of RAD52 inhibition in killing BRCA-deficient cancer.”

Comment: RNF168 page 13 : why restoring HR in BRCA1 deficient tumor would improve cancer therapy?

Answer: I must have misinterpreted the conclusion of Munoz et al. I have changed the sentence:

“The authors postulated that RNF168 may be important in diagnosis of tumors with DSBR deficiency and this protein can be considered in therapy of BRCA1-deficinet tumors to restore HRR.”

into:

“The authors postulated that RNF168 may be important in diagnosis of tumors with DSBR deficiency and this protein can be considered in therapy of BRCA1-deficinet tumors to prevent HDR restoration.”

Comment: Diallyl disulfide (DDS, page 14) inhibits resection and thus HR repair, not only SSA. This information is not relevant here.

Answer: Absolutely! The more that it concerns yeast. I have removed the fragment:

“Combining chemo- or radiotherapy with natural compounds may result in better outcomes of cancer treatment [97]. Using a yeast SSA model, Kuo et al. showed that diallyl disulfide (DDS) inhibited DNA repair and sensitized SSA cells to a single DSB [98]. They showed that the effect they observed resulted from downregulation of end resection proteins Sae2 (SUMO-activating Enzyme Subunit 2) and Exo1 (Exonuclease 1). These authors also showed that DDS prevented recruiting the MRX (Mre11-Rad50-Xrs2) and Mec1-Dcd2 complexes to a DSB. Therefore, DDS inhibited activation of the G2/M checkpoint in a way dependent on the homologs of mammalian ATM and ATR proteins and could assist cancer therapy with DSB-inducing factors.”

Comment: Conclusion page 17. SSA do not contribute to HR-deficient cell death. On the contrary, SSA inhibition induce synthetic lethality in these genetic background.

Answer: I have changed the sentence:

“It may also contribute to the death of cancer cells deficient in other DSBR pathway, especially HRR with impaired RAD51 or BRCA1/2.”

into:

“Its inhibition may also contribute to the death of cancer cells deficient in other DSBR pathway, especially HRR with impaired RAD51 or BRCA1/2.”

Comment: The manuscript requires extensive proofreading.

Answer: I have done my best to correct all errors and mistakes.

Round 2

Reviewer 2 Report

This revised version has been significantly improved. Misleading informations have globally been corrected according to previous comments.

I still have few minor comments:

New Figure 1: MMEJ does not require extensive resection, as explained by the author (page 7, last two lines). Thus, does author make a difference between B-NHEJ after initial end resection (first 100 nucleotides), and MMEJ that is shown after extensive resection? To me, the figure should discriminate between B-NHEJ and extensive resection (DNA2 and EXO1 dependent) just after initial resection, and then discriminate between RAD51 and no RAD51 after extensive resection.

Comment on S. cerevisiae RAD52 that has not been adressed. Author writes (p15 in the new version): "Although it is not completely clear how RAD52 functions in HRR in connection with RAD51, studies in S. cerevisiae suggest that RAD52 may assist to load RAD51 onto ssDNA at DSBs and compete with SSB proteins (new ref 82)." It is well established that S. cerevisiae RAD52 displaces RPA, but human RAD52 cannot (HR mediator role). Instead, BRCA2 and PALB2 (absent in S. cerevisiae) make this job in vertebrates. This is well explained in Wright et al, JBC 2019: "In yeast, Rad52 mediates the replacement of RPA with Rad51 in a mechanism where ssDNA wraps around Rad52, destabilizing the RPA-ssDNA interaction while promoting Rad51 binding (HR mediator function) through physical interaction between Rad51 and Rad52. Yeast Rad52 defines its epistasis group of HR proteins, being necessary for all Rad51 filament formation in vivo. Interestingly, though RAD52 is conserved as a protein, its dominant mediator function is not." To conclude, author cannot refer to S. cerevisiae RAD52 mediator function to explain the role of human RAD52 in HRR.

page 15 : That study may contribute to understanting mechanisms of oncogenesis...

Author Response

This revised version has been significantly improved. Misleading informations have globally been corrected according to previous comments.

I still have few minor comments:

Comment: New Figure 1: MMEJ does not require extensive resection, as explained by the author (page 7, last two lines). Thus, does author make a difference between B-NHEJ after initial end resection (first 100 nucleotides), and MMEJ that is shown after extensive resection? To me, the figure should discriminate between B-NHEJ and extensive resection (DNA2 and EXO1 dependent) just after initial resection, and then discriminate between RAD51 and no RAD51 after extensive resection.

Answer: I have corrected the figure according to that comment. However, to follow it I had to skip length of homology and synapsis formation not to complicate the figure too much. I have modified the figure legend and corresponding modifications have been also introduced into the main text illustrated by the figure.

Comment: Comment on S. cerevisiae RAD52 that has not been adressed. Author writes (p15 in the new version): "Although it is not completely clear how RAD52 functions in HRR in connection with RAD51, studies in S. cerevisiae suggest that RAD52 may assist to load RAD51 onto ssDNA at DSBs and compete with SSB proteins (new ref 82)." It is well established that S. cerevisiae RAD52 displaces RPA, but human RAD52 cannot (HR mediator role). Instead, BRCA2 and PALB2 (absent in S. cerevisiae) make this job in vertebrates. This is well explained in Wright et al, JBC 2019: "In yeast, Rad52 mediates the replacement of RPA with Rad51 in a mechanism where ssDNA wraps around Rad52, destabilizing the RPA-ssDNA interaction while promoting Rad51 binding (HR mediator function) through physical interaction between Rad51 and Rad52. Yeast Rad52 defines its epistasis group of HR proteins, being necessary for all Rad51 filament formation in vivo. Interestingly, though RAD52 is conserved as a protein, its dominant mediator function is not." To conclude, author cannot refer to S. cerevisiae RAD52 mediator function to explain the role of human RAD52 in HRR.

Answer: Therefore, I have removed the sentence:

“Although it is not completely clear how RAD52 functions in HRR in connection with RAD51, studies in S. cerevisiae suggest that RAD52 may assist to load RAD51 onto ssDNA at DSBs and compete with SSB proteins [82].”

as inappropriate speculation.

Comment: page 15 : That study may contribute to understanting mechanisms of oncogenesis...

Answer: I have corrected that.